# Biodegradation of Tetrahydrofuran by the Newly Isolated Filamentous Fungus *Pseudallescheria boydii* ZM01

**DOI:** 10.3390/microorganisms8081190

**Published:** 2020-08-05

**Authors:** Hao Ren, Hanbo Li, Haixia Wang, Hui Huang, Zhenmei Lu

**Affiliations:** MOE Laboratory of Biosystem Homeostasis and Protection, College of Life Sciences, Zhejiang University, Hangzhou 310058, China; 11707034@zju.edu.cn (H.R.); 21707029@zju.edu.cn (H.L.); 11207031@zju.edu.cn (H.W.); huanghuilengyue@163.com (H.H.)

**Keywords:** tetrahydrofuran, *Pseudallescheria boydii*, 2-hydroxytetrahydrofuran, biodegradation

## Abstract

Tetrahydrofuran (THF) is widely used as a precursor for polymer syntheses and a versatile solvent in industries. THF is an environmental hazard and carcinogenic to humans. In the present study, a new THF-degrading filamentous fungus, *Pseudallescheria boydii* ZM01, was isolated and characterized. Strain ZM01 can tolerate a maximum THF concentration of 260 mM and can completely degrade 5 mM THF in 48 h, with a maximum THF degradation rate of 133.40 mg THF h^−1^ g^−1^ dry weight. Growth inhibition was not observed when the initial THF concentration was below 150 mM, and the maximum THF degradation rate was still maintained at 118.21 mg THF h^−1^ g^−1^ dry weight at 50 mM THF, indicating the great potential of this strain to degrade THF at high concentrations. The initial key metabolic intermediate 2-hydroxytetrahydrofuran was detected and identified by gas chromatography (GC) analyses for the first time during the THF degradation process. Analyses of the effects of initial pH, incubation temperature, and heavy metal ions on THF degradation revealed that strain ZM01 can degrade THF under a relatively wide range of conditions and has good degradation ability under low pH and Cu^2+^ stress, suggesting its adaptability and applicability for industrial wastewater treatment.

## 1. Introduction

Tetrahydrofuran (THF) is a colorless, highly soluble, and volatile heterocyclic compound that is widely used as an important basic feedstock for chemical syntheses and is an excellent solvent for lacquers and printing inks [1,2]. With the increasing consumption of THF, large amounts of industrial wastewater containing THF have been released into groundwater, rivers, and lakes [3,4]. Furthermore, increased attention has been paid to the carcinogenic effects of THF-polluted groundwater on human health after its toxicological effects were thoroughly demonstrated by a variety of assays in vivo. Various studies in female mice have demonstrated that THF can enhance tumor formation by inducing cell proliferation, can cause nervous system dysfunction, and can induce DNA damage [5,6,7]. Considering the advantages of microbial degradation, including the safety of disposal and cost effectiveness, the key to refractory organic pollutant treatment will be the exploitation of microbes from the natural environment with highly efficient pollutant-degrading activities [8,9]. THF is also classified as a refractory pollutant due to the high bond energy obstruction of C–O (360 KJ mol^−1^) [10]. A number of bacteria, including gram-positive *Rhodococcus* sp., *Pseudonocardia* sp., and *Mycobacterium* sp. and gram-negative *Flavobacterium* sp. and *Pseudomonas* sp. have been reported to aerobically grow on THF [11,12,13,14,15]. Although many THF-degrading bacteria have been reported, most of them cannot utilize THF at concentrations higher than 35 mM, and the lag phase is prolonged with increased initial substrate concentrations. The widely reported THF-degrading strain *Rhodococcus ruber* YYL showed the highest THF tolerance concentration of up to 200 mM among all the reported THF-degrading strains. Strain YYL can completely degrade 6 mM THF within 32 h without a lag phase, whereas the lag phase increased to more than 96 h when the initial concentration of THF was increased to 100 mM [16]. *R. aetherivorans* M8 exhibits an obvious lag phase of 5 days in the presence of 15 mM THF, which was prolonged to more than 15 d when the THF concentration was increased up to 30 mM [17]. *Pseudomonas oleovorans* DT4, which has a superior maximum THF degradation rate compared to the other reported THF-degrading strains, cannot mineralize THF completely in the presence of THF concentrations higher than 30 mM, with the degradation ratio decreasing to 45% with 40 mM THF [1]. Therefore, growth inhibition and incomplete degradation are major issues with respect to the ability of bacteria to efficiently degrade THF when under the stress of high substrate concentrations.

Filamentous fungi are involved in multiple pathways associated with wastewater treatment, such as organic compound biodegradation, the formation of sludge flocs, settleability of the treated sludge, and sludge detoxification [18]. Filamentous fungi grow as thread-like hyphae and reproduce by sporulation [19]. The hyphae can immobilize sludge particles, resulting in the formation of pellets, which leads to microbial aggregation as natural flocculants [20,21]. Notably, filamentous fungi can also improve the dewaterability and filterability of sludge by modifying its porosity structure [22]. Many studies have shown that compounds that are highly toxic toward bacteria may be harmless to fungi, such as antibiotics [21]. In addition, the high contents of functional groups in the cell walls of fungi, such as amino, amide, and phosphate groups, promote the ability of fungi to absorb low concentrations of heavy metals as micronutrient sources and to resist heavy metal stress [23]. The ability of filamentous fungi to form mycelial networks, to undergo detoxification, and to maintain stress resistance and richness in terms of biodiversity enhances their potential application in bioremediation [24,25]. To date, only three fungi, including *Aureobasidium pullulans*, *Cordyceps sinensis* [26], and *Graphium* sp. ATCC 58400 [27] have been reported to grow on THF, and the THF degradation characteristics of fungi have rarely been evaluated. Thus, studies on the THF degradation characteristics of THF-degrading fungi is important for developing bioremediation technologies for THF-containing wastewater.

Two different microbial THF degradation pathways have been proposed in previous studies, but some metabolites remain speculative, such as 2-hydroxytetrahydrofuran. In the oxidation pathway, THF is inferred to be initially oxidized to 2-hydroxytetrahydrofuran, and 2-hydroxytetrahydrofuran is then oxidized to γ-butyrolactone before being transformed to 4-hydroxybutyrate. 2-hydroxytetrahydrofuran is a cyclic hemiacetal that can also autonomously form its tautomer 4-hydroxybutyraldehyde, which can be directly converted to 4-hydroxybutyrate. The metabolite 4-hydroxybutyrate, which is common to both metabolic pathways, is further oxidized to succinate and then thoroughly mineralized in the tricarboxylic acid cycle [17,28,29]. Several THF degradation intermediates in the proposed THF degradation pathway have been detected and verified in previous studies. Small amounts of γ-butyrolactone were detected as an intermediate in *Graphium* sp. ATCC 58400, *Mycobacterium vaccae* JOB-5, and *Rhodococcus* sp. DTB after incubation with THF [27,29,30]. In addition, the derivatized product of 4-hydroxybutyrate was finally detected and confirmed by gas chromatography-mass spectrometry (GC–MS) in the resting cell reaction of *R. aetherivorans* M8, which was initiated by the addition of 5 mM THF [17]. However, the initial key monooxygenation metabolite in both degradation pathways, 2-hydroxytetrahydrofuran, has not been detected and confirmed during the THF degradation process in all reported THF-degrading strains thus far [31,32].

In the present study, a new filamentous fungus capable of sustained growth on THF as the sole carbon source was isolated and identified. We aimed to characterize the potential of *Pseudallescheria boydii* ZM01 in THF degradation under different culture conditions and to further evaluate the range of substrates that strain ZM01 can degrade. The metabolic intermediates during THF degradation were also identified, and the metabolic pathway of THF in strain ZM01 was proposed.

## 2. Materials and Methods

### 2.1. Chemicals, Media, and Growth Conditions

THF (>99% in purity), γ-butyrolactone (>99% in purity), 1,4-dioxane (>99% in purity), and 1,4-butanediol (>98% in purity) were purchased from Aladdin Industrial Corporation (Shanghai, China) while 2-hydroxytetrahydrofuran (>95% in purity) was purchased from Macklin Biochemical Co., Ltd. (Shanghai, China). All other chemicals used as degradation substrates were of reagent or analytical grade.

Minimal salt medium (MSM) supplemented with THF was initially used to isolate and culture THF-degrading microorganisms. MSM was prepared with the following components (per liter): 4.5 g Na_2_HPO_4_∙12H_2_O, 1.0 g KH_2_PO_4_, 1.5 g NH_4_Cl, 0.2 g MgSO_4_∙7H_2_O, 0.03 g CaCl_2_∙2H_2_O, and 1 mL trace element medium stock solution (per liter: 1.0 g FeSO_4_∙7H_2_O, 0.02 g CuSO_4_∙5H_2_O, 0.014 g H_3_BO_3_, 0.1 g MnSO_4_∙4H_2_O, 0.1 g ZnSO_4_∙7H_2_O, 0.02 g Na_2_MoO_4_∙2H_2_O, and 0.02 g CoCl_2_∙6H_2_O). Mycelia of filamentous fungi were inoculated onto potato dextrose agar (PDA) plates at 30 °C for 10 days to produce conidia. Conidia were harvested by washing the cultured PDA plates with 5 mL of distilled water and were counted with a hemocytometer as previously described [33]. Then, the conidial suspensions were transferred into 100 mL of MSM (final concentration of 2 × 10^6^ conidia mL^−1^) for THF degradation characterizations and metabolic intermediates analyses. All liquid culture experiments were performed in 250-mL narrow-mouthed shake flasks, and the bottle mouth was sealed to reduce the evaporation of THF during the cultivation process.

### 2.2. Isolation and Identification of the THF-Degrading Fungus

The THF-degrading strain was enriched and isolated from activated sludge collected from Jiaxing Chemical Factory, Zhejiang Province, China. The collected samples were acclimatized in 100 mL of MSM supplemented with different concentrations of THF from 20 mM to 50 mM at 30 °C with continuous shaking at 200 rpm. After 28 days of continuous passage cultivation, the stable degradation consortium was serially diluted and plated on MSM agar plates containing 20 mM THF at 30 °C for 7 days. The colonies were continuously selected, and pure cultures were finally isolated after continuous passage cultivation for more than five generations. Fungal genomic DNA was extracted from mycelia using an Omega Fungal DNA kit (Norcross, GA, USA). The internal transcribed spacer (ITS) region of strain ZM01 was amplified using the universal primers and sequenced by TsingKe Biotech (Hangzhou, China). The obtained ITS region was aligned to sequences of other related species using the program BLAST (https://blast.ncbi.nlm.gov). All the reference sequences were aligned using Clustal X, and a phylogenetic tree was generated with MEGA6 using the neighbor-joining method [34].

### 2.3. Biodegradation of THF by Strain ZM01

Biological tolerance analyses of strain ZM01 were conducted in 100 mL of MSM at 30 °C with initial THF concentrations ranging from 5 to 300 mM in a rotary shaker (200 rpm). Samples collected at designated intervals were analyzed by GC to assess the residual THF concentration. The mycelia were filtered from fermentation liquid with gauze and vacuum freeze-dried to measure dry weight. The effects of THF volatilization have been considered and evaluated in our previous studies [16,35], and abiotic control was prepared to confirm the degradation by strain ZM01 in all experimental designs.

The Monod equation was unsatisfactory to determine the kinetic parameters due to substrate inhibition, while Haldane’s growth model was well-fitted even at inhibitory levels of the substrate. Thus, Equation (1) was adopted and is shown below [36]:(1)$$\mu =\frac{\mu_{max}S}{K_{S}+S+\left(\frac{S^{2}}{K_{i}}\right)}$$
where *μ_max_* is the maximum specific growth rate (h^−1^), *S* is the substrate concentration (mg/L), *K_s_* is the half-saturation constant (mg/L), and *K_i_* is the inhibition constant (mg/L). Thus, the specific degradation rate (*v*) can be calculated using the specific growth rate (*μ*) and growth yield (*Y_x/s_*) as below [37]:(2)$$v=\frac{d_{s}}{d_{t}}\;=-\;\frac{\mu }{Y_{X/S}}$$

The THF degradation ratio was calculated as follows: THF degradation ratio (%) = $\frac{C_{0}-C_{t}}{C_{0}}$ × 100 (3), where *C_o_* is the initial concentration of THF and *C_t_* is the residual THF concentration in fermentation liquid.

### 2.4. Effects of Different Factors on THF Biodegradation of Strain ZM01

In single factor experiments, conidial suspensions of strain ZM01 were incubated in 100 mL MSM containing 50 mM THF with shaking at 200 rpm. Various environmental factors were assayed to investigate their effects on THF degradation characteristics, including pH (4.0, 5.0, 6.0, 7.0, 8.0, 9.0, 10.0, and 11.0); temperature (15, 20, 25, 30, 35, and 40 °C); inoculum size (5 × 10^5^, 10^6^, 2 × 10^6^, 3 × 10^6^, and 4 × 10^6^ conidia mL^−1^); Cd^2+^ (0.05, 0.1, 0.3, 0.5, and 1 mM); Pb^2+^ (0.1, 0.5, 1, 2 and 3 mM); Mn^2+^ (1, 2, 3, 4, and 5 mM), and Cu^2+^ (0.5, 1, 2, 3, and 4 mM). The mother conidial suspensions with an initial concentration of 10^8^ conidia mL^−1^ were prepared to analyze the effect of inoculum size on THF degradation. The effects of different culture conditions were finally evaluated based on the specific THF degradation ratio.

### 2.5. Utilization of Different Substrates by Strain ZM01

Various substrates were selected to evaluate the degradation potential of strain ZM01, including cyclic aliphatic ethers, benzene homologs, alkanes, alcohols, ketones, and other THF-related compounds. Batch experiments were performed in 100 mL of MSM for 5 days, and the initial concentration of each substrate was 20 mM. The mycelia were collected and finally measured using the methods described previously.

### 2.6. Detection and Identification of THF and Related Metabolites

The detection of the THF concentration was conducted as described in a previous study [38]. The THF concentration was measured by GC-2014C gas chromatography equipped with a flame ionization detector (FID) and an AOC-20i Auto injector (SHIMADZU, Shanghai, China). All samples were centrifuged at 10,000 rpm for 2 min, and the supernatant was subsequently collected after being filtered through a 0.45-μm filter for pretreatment. Samples at different time points corresponding to the growth curve were taken to detect the intermediate metabolites. The detection program was as follows: the initial temperature was set at 60 °C, gradually increased to 160 °C at a rate of 20 °C min^−1^, and then held at 160 °C for 5 min. The reference standards, such as THF, 2-hydroxytetrahydrofuran, and γ-butyrolactone, were appropriately prepared at different concentrations to generate standard curves for use in intermediate identification and concentration calculations.

To identify the intermediate metabolites of THF produced by strain ZM01, the intermediate metabolites were concentrated by organic solvent extraction. To this end, 20 mL of ethyl acetate was added directly to 70 mL of fermentation liquid, and the mixture was vigorously shaken for 20 min to completely extract the metabolites. The extraction phase was then evaporated to dryness and dissolved in 1 mL of ethyl acetate for GC-MS analysis. The final concentrated metabolic intermediates were identified using a GC-MS instrument (Agilent 7890B-7000C, Shanghai, China) equipped with an HP-5ms capillary column (30 m × 0.25 mm × 0.25 μm). The column oven temperature was maintained at 40 °C for 3 min and then increased to 120 °C for 2 min at a rate of 10 °C min^−1^. The eluted compounds were ionized by electron impact (70 eV) after passing through a GC-MS interface maintained at 250 °C. Ions with masses ranging from 10 to 400 atomic mass units were scanned at 1-s intervals. Compounds were identified by comparing their fragmentation patterns with those of the authentic standards and the mass spectra library (National Institutes of Standards Technology, Gaithersburg, MD, USA).

## 3. Results and Discussion

### 3.1. Isolation and Identification of the THF-Degrading Fungus

A filamentous fungus capable of growing in MSM supplemented with THF as the sole carbon source was successfully isolated from activated sludge by enrichment cultivation and designated strain ZM01. Strain ZM01 could grow as white thread-like hyphae and could develop conidia growing directly on vegetative hyphae after cultivation for 7 d at 30 °C. Most conidia were usually brown, thick-walled, globose to subglobose. The shape of conidiogenous cells was cylindrical, and no yellow diffusible pigment was produced during the cultivation of strain ZM01 on PDA at 30 °C (Appendix A). The microscopic characteristics of strain ZM01 were consistent with the typical microscopic features of *Pseudallescheria* sp. Based on phylogenetic analysis of the ITS region sequences (GenBank Accession No. MT754398) (Figure 1), strain ZM01 was observed to be closely related to *P. boydii* (GenBank Accession No. AY213683) with a high degree of similarity (99%). Thus, on the basis of morphological characteristics and phylogenetic relationships analysis, we identified the isolated THF-degrading fungus as *Pseudallescheria boydii* strain ZM01. To the best of our knowledge, this is the first report showing that THF can be metabolized by a *Pseudallescheria* sp. strain.

### 3.2. Degradation Characteristics of Strain ZM01

The substrate tolerance of strain ZM01 was studied at various THF concentrations. As shown in Figure 2, strain ZM01 could grow in the presence of THF at concentrations ranging from 5–260 mM, while further increasing the THF concentration to 300 mM resulted in complete inhibition of THF utilization by strain ZM01. Thus, the maximum tolerable THF concentration of strain ZM01 was determined to be 260 mM, which is higher than that of *R. ruber* YYL (200 mM), *P. oleovorans* DT4 (100 mM), *R. aetherivorans* M8 (35 mM), and *Pseudonocardia* sp. K1 (60 mM) [1,15,17,30]. As the growth and degradation characteristics of previously evaluated THF-degrading strains were determined at low THF concentrations, we assessed the ability of strain ZM01 to degrade 5 mM of THF. The results showed that strain ZM01 could completely mineralize 5 mM THF in 48 h, with a maximum specific growth rate (*μ_max_*) of 0.23 h^−1^. The maximum THF degradation rate of strain ZM01 at 5 mM THF was 133.40 mg THF h^−1^ g^−1^ dry weight at a yield (*Y_X/C_*) of up to 0.058 (dry weight substrate carbon^−1^) (Figure 2), which was as high as that reported for *R. ruber* YYL (137.60 mg THF h^−1^ g^−1^ dry weight) [16] but lower than reported for *P. oleovorans* DT4 (203.9 mg THF h^−1^ g^−1^ dry weight) [1]. Considering the different growth and reproduction patterns between fungi and bacteria, a comparative analysis of the degradation ability of all the reported THF-degrading fungi was also conducted. Strain ZM01 showed the highest THF degradation rate among fungi, and the maximum THF degradation rate of strain ZM01 was almost twelve-fold higher than that of the representative THF-degrading fungus *Graphium* sp. strain ATCC 58400 (11.22 mg THF h^−1^ g^−1^ dry weight) [27]. Notably, the maximum degradation rate observed for *P. oleovorans* DT4 significantly decreased to only 22.64 mg THF h^−1^ g^−1^ dry weight when the substrate concentration was increased to 20 mM, while it was 118.21 mg THF h^−1^ g^−1^ dry weight at 50 mM THF in strain ZM01, indicating that the degradation ability of strain ZM01 is superior to that of *P. oleovorans* DT4 at high THF concentrations.

Incomplete degradation and a prolonged lag phase at higher substrate concentrations are frequently encountered problems for most THF-degrading bacteria. The results of our investigation showed that strain ZM01 could completely utilize 20 mM THF in 120 h and 50 mM THF in 240 h at 30 °C, with no lag phase observed when the initial THF concentration was below 150 mM (Figure 2). Compared with the representative THF-degrading bacteria *R. ruber* YYL and *P. oleovorans* DT4, which were unable to completely mineralize THF higher than 30 mM and experienced a prolonged lag phase (more than 70 h) at THF concentrations below 50 mM, strain ZM01 exhibits superior THF degradation and growth characteristics under substrate stress [1,16]. The high THF tolerance, no lag phase under high THF stress, and high THF degradation rate observed for strain ZM01 indicate that it has great potential for THF removal in industrial wastewater. In addition, the ability of filamentous fungi to form large and strong flocs, to detoxify toxic organic compounds, and to produce nonspecific substrate enzymes enables them to successfully colonize a microbial consortium and to effectively degrade pollutants in a bioaugmentation treatment system [21,39,40].

### 3.3. Effects of pH, Temperature, and Inoculum Size on THF Biodegradation

The effects of pH, temperature, and inoculum size were tested over a wide range and evaluated based on the THF degradation ratio (Figure 3). The pH value was determined to be a key factor influencing THF degradation by strain ZM01. Acidification often occurs during THF degradation along with the production of some acidic metabolites, such as succinic acid, pyruvic acid, citramalic acid, and acetoacetic acid [27,35]. Strain ZM01 was capable of degrading THF at a wide pH range from 4.0 to 11.0, and an increase in the initial pH from 4.0 to 7.0 resulted in a gradual increase in the THF degradation ratio (Figure 3a). Studies on the degradation ability of THF-degrading strains at low pH were rare, and only five THF-degrading strains, including *Afipia* sp. D1, *Mycobacterium* sp. D6, *Pseudonocardia* sp. D17, and *Rhodococcus* sp. M8 and JCM 14343 have been reported to grow at initial pH below 6.0 when using THF or 1,4-dioxane as the substrate [11,17,41]. When strain ZM01 was cultivated at pH 4.0, the THF degradation ratio was maintained at 64.36%, which is almost 76% of the highest degradation ratio observed at pH 7.0. Compared with *Afipia* sp. D1 and *Rhodococcus* sp. JCM14343, which maintained 50 and 56% of the highest 1,4-dioxane degradation ratio at pH 4.0, respectively, strain ZM01 has a higher tolerance to acidic environments [11,41]. THF degradation by *R. ruber* YYL was inhibited at pH values lower than 7.0, and the degradation ratio was reduced by 40% when the initial pH value decreased from 8.26 to 7.0 [16,38]. Notably, although the pH decreased from 4.0 to 3.0 after strain ZM01 was cultivated for 240 h (Appendix A), the THF degradation ratio at the initial pH 4.0 was only 20% lower than that observed at the optimal pH 7.0 (Figure 3a). These results demonstrate that strain ZM01 has an excellent degradation ability under low pH stress, as indicated by its activity over a wide pH range.

The influence of the temperature used to cultivate strain ZM01 on the degradation ratio of THF is shown in Figure 3b. At temperatures either above or below the optimal degradation temperature, the THF degradation ratio of strain ZM01 decreased. The highest degradation ratio, close to 85%, was observed at 30 °C. However, THF degradation was also observed at the extreme temperatures of 15 and 40 °C, and the degradation ratios at these temperatures remained at 25% of that observed at the highest degradation ratio. Compared to *R. aetherivorans* strain M8, which cannot grow at 22 °C [17], strain ZM01 could degrade THF over a relatively broad range of temperatures. Figure 3c shows the effects of inoculum size on the THF degradation ratio. When the inoculum size increased from 5 × 10^5^ to 2 × 10^6^ conidia mL^−1^, the THF degradation ratio increased from 48.54% to 85.71%. However, the degradation ratio was no longer significantly improved when the inoculum size was more than 2 × 10^6^ conidia mL^−1^ due to the limited availability of some factors, such as trace elements and THF degradation-related monooxygenases [28,42]. Based on the previous results, an inoculum size of 2 × 10^6^ conidia mL^−1^ was used in subsequent experiments.

### 3.4. Effects of Metal Ions on THF Degradation

Considering the toxic effects of heavy metals, many studies have investigated the influence of heavy metals on the biodegradation of pollutants by pure cultures or microbial consortia [43,44]. We determined the tolerance of strain ZM01 to different types of metal ions at various concentrations (Figure 4). Overall, the THF degradation ratios of strain ZM01 decreased with increasing metal ions concentrations. The inhibitory effect of metal ions at a concentration of 1.0 mM on THF degradation was in the following order: Cd^2+^ > Cu^2+^ > Pb^2+^ > Mn^2+^. This trend was similar to that observed in a previous study on the effects of metal ions on 1,4-dioxane degradation by *P. dioxanivorans* CB1190 [45]. Among the metal ions assayed, strain ZM01 was the most sensitive to Cd^2+^, and the THF degradation ratio decreased from 63.9% to 30.7% when the Cd^2+^ concentration increased from 0.05 to 1 mM. The slightly stimulating effect of a small amount of Mn^2+^ (less than 2 mM) on THF degradation was evaluated, and the THF degradation ratio decreased by 24.6% with the addition of 4 mM Mn^2+^. A THF degradation ratio of more than 50% was observed when the cultures contained 1 mM Pb^2+^ and Cu^2+^, and further increasing the concentrations of these metal ions resulted in a reduction in THF biodegradation.

In previous studies, Cu^2+^ was shown to have a strong inhibitory effect on THF degradation and was the most toxic trace element among Co^2+^, Cu^2+^, Ca^2+^, Zn^2+^, Mg^2+^, and Fe^2+^ [16]. A study on the bioavailability and toxicity of Cu^2+^ showed that the addition of more than 10 mg/L Cu^2+^ caused a prolonged initial lag phase and decreased the 1,4-dioxane degradation rate [45,46]. Additionally, strain *P. dioxanivorans* CB1190 was shown to degrade 120 mg/L 1,4-dioxane with a degradation rate of 30.23 ± 2.86 mg L^−1^ d^−1^, which decreased by nearly 90% with the addition of 0.15 and 0.3 mM Cu^2+^ [45]. In addition, Cu^2+^ was also observed to sharply inhibit the THF degradation by consortium H1, with a THF degradation ratio of less than 10% observed after 108 h when the Cu^2+^ concentration was increased to 1 mM [47]. However, the THF degradation ratio was maintained at 51.3% and was only 12% lower than that of the normal control group when strain ZM01 was cultivated in the presence of 1 mM Cu^2+^, indicating that it has better THF degradation potential under Cu^2+^ stress.

### 3.5. Degradation of Different Substrates by Strain ZM01

Strain ZM01 was capable of utilizing a variety of substrates (Table 1), including cyclic aliphatic ethers (2-hydroxytetrahydrofuran, 3-hydroxytetrahydrofuran, and dimethylfuran), benzene, ethyl acetate, *n*-hexane, γ-butyrolactone, and 1,4-butanediol. In addition to THF, higher biomass was also observed in the presence of the candidate metabolites 2-hydroxytetrahydrofuran (58.9 ± 2.1 mg), γ-butyrolactone (53.7 ± 1.5 mg), and 1,4-butanediol (71.7 ± 1.5 mg), which is consistent with the results reported for THF-degrading bacteria [17,30]. The degradation of γ-butyrolactone was accelerated when THF and γ-butyrolactone were used as substrates (Appendix A). Similar to other reported THF-degrading strains, such as *P. dioxanivorans* CB1190, *Afipia* sp. D1, and *Pseudonocardia* sp. strain ENV478, strain ZM01 could also utilize 1,4-dioxane as the sole carbon and energy source in pure culture [11,48,49]. 1,4-Dioxane can also be co-metabolically oxidized by some aerobic microorganisms grown on THF [15,37], which may be due to the analogous structure of THF and 1,4-dioxane, and the degradation of dioxane and THF may be initiated by the same monooxygenase in these bacteria [50].

### 3.6. Proposed THF Degradation Pathway in Strain ZM01

Although various methods have been used in an attempt to elucidate the pathway involved in THF metabolism in previous investigations, such as resting cell reaction, the metabolite 2-hydroxytetrahydrofuran still has not been detected and identified yet. In the present study, the metabolic intermediates of THF were measured and identified at different cultivation periods of strain ZM01 when utilizing 20 and 50 mM THF as substrates. As shown in Figure 5b, three peaks that appeared at retention times of 1.807, 5.605, and 6.952 min were detected during 20 mM THF degradation by strain ZM01. According to the retention times of the reference standards shown in Figure 5a, they were identified as THF, 2-hydroxytetrahydrofuran, and γ-butyrolactone, respectively. During the initial stage of THF degradation, spores germinate to form hyphae and the metabolites are slowly utilized, providing a good opportunity for the accumulation of metabolic intermediates. The key metabolite 2-hydroxytetrahydrofuran (retention time of 5.605 min) first appeared at 21 h and accumulated to a maximum concentration of 0.129 mM after cultivation for 30 h (Figure 5c). There was a brief accumulation of 2-hydroxytetrahydrofuran during the degradation of 20 and 50 mM THF, and the earliest time point at which 2-hydroxytetrahydrofuran detected was delayed to 36 h after increasing the THF concentration to 50 mM (Appendix A). However, no significant increase was observed in the maximum concentration of 2-hydroxytetrahydrofuran after increasing the substrate concentration to 50 mM (only 0.107 mM). According to our investigation, the reasons why the key metabolite 2-hydroxytetrahydrofuran failed to be detected during THF degradation by bacteria in previous reports may be attributed to (i) the low concentration of intermediates during metabolism, (ii) the fast conversion of intermediates, and (iii) the different growth patterns between fungi and bacteria.

With the consumption of THF, the concentration of γ-butyrolactone increased proportionally over 84 h and then decreased gradually during THF degradation (Figure 5c). The maximum amount of γ-butyrolactone (0.758 mM) was detected after 84 h when 84% of the total THF was consumed, which was almost five times that observed using *Graphium* sp. strain ATCC 58400 [27]. The fermentation supernatants after cultivation for 30 and 84 h by strain ZM01 when using 20 mM THF as substrate were collected and concentrated for GC-MS analysis. The retention time of one peak at 7.153 min was finally identified by GC-MS as γ-butyrolactone (Figure 6a). No other THF-derived metabolites, including the tautomer 4-hydroxybutyraldehyde, were detected during the cultivation processes. Based on the results described above and previously reported data, we propose that THF is degraded via the oxidation pathway in *P. boydii* ZM01 (Figure 6b).

## 4. Conclusions

The THF-degradation potential of a new filamentous fungus, *P. boydii* ZM01, was evaluated in this study. Strain ZM01 could tolerate THF at concentrations of up to 260 mM, which is superior to that of other reported THF-degrading strains. The maximum THF degradation rate for strain ZM01 was 133.40 mg THF h^−1^ g^−1^ dry weight at 5 mM THF and remained at 118.21 mg THF h^−1^ g^−1^ dry weight when using 50 mM THF as a substrate, indicating the great potential of strain ZM01 for the treatment of industrial wastewater contaminated with high concentrations of THF. The effects of initial pH, culture temperature, and heavy metal ions on THF degradation by strain ZM01 revealed that it has a wide range of growth and good degradation ability under low pH and Cu^2+^ stress. In addition to THF, strain ZM01 can also utilize 1,4-dioxane, dimethylfuran, benzene, n-hexane, and ethyl acetate as growth substrates. The results also showed that strain ZM01 degrades THF through the oxidation pathway, and notably, the THF-derived metabolite, 2-hydroxytetrahydrofuran, was detected and confirmed during THF degradation for the first time.

## Figures and Tables

**Figure 1 microorganisms-08-01190-f001:**
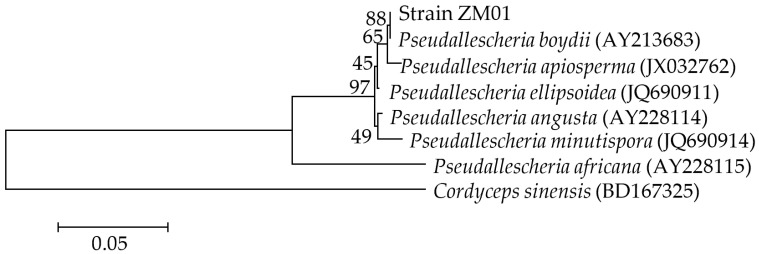
The phylogenetic dendrogram for strain ZM01 and closely related *Pseudallescheria* strains based on the internal transcribed spacer (ITS) region sequences: The representative tetrahydrofuran (THF)-degrading fungus *C. sinensis* is used as the root. Numbers after the names of organisms are accession numbers of published sequences. Numbers adjacent to the branches indicate the bootstrap values based on 1000 replicates.

**Figure 2 microorganisms-08-01190-f002:**
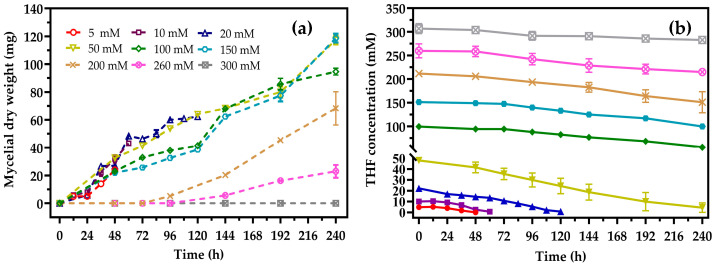
Growth (**a**) and THF degradation (**b**) of strain ZM01 with different initial THF concentrations at 30 °C: All values represent the mean of three independent biological replicates.

**Figure 3 microorganisms-08-01190-f003:**
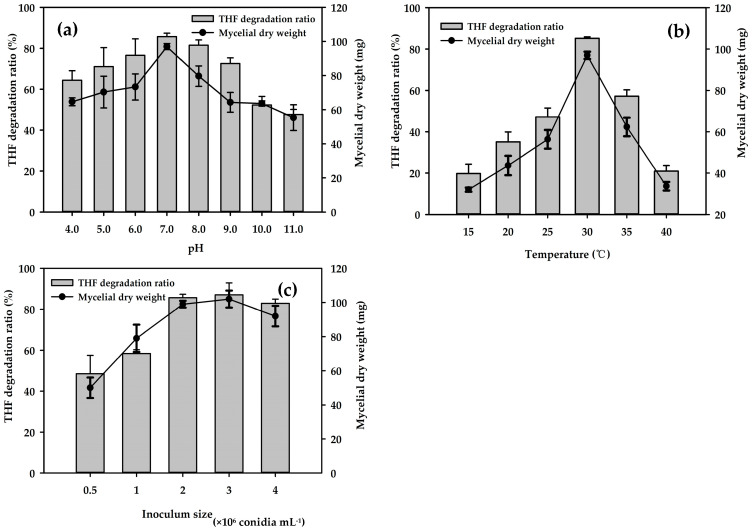
Effect of different initial pH values, incubation temperatures, and inoculum sizes on the biodegradation of THF by strain ZM01: The THF degradation ratio was detected after cultivation for 8 d at 50 mM THF. (**a**) pH; (**b**) temperature; and (**c**) inoculum size.

**Figure 4 microorganisms-08-01190-f004:**
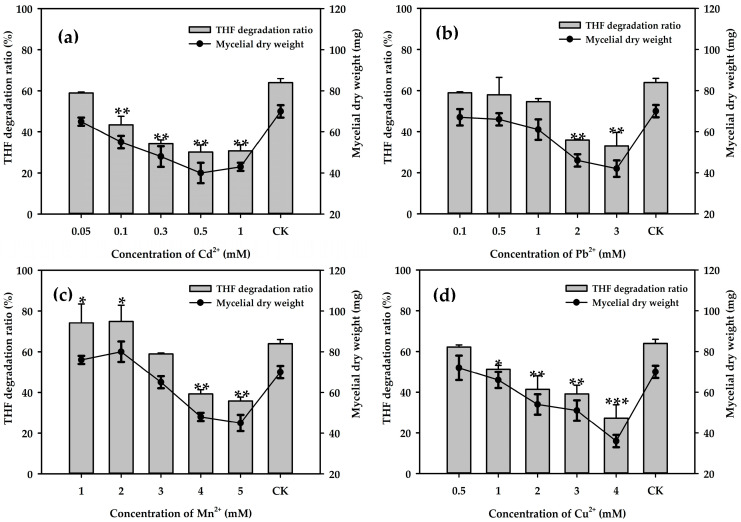
The effect of various concentrations of heavy metal ions on the biodegradation of THF by strain ZM01: The THF degradation ratio was detected after cultivation for 6 d at 50 mM THF. (**a**) Cd^2+^; (**b**) Pb^2+^; (**c**) Mn^2+^; and (**d**) Cu^2+^. Significance was analyzed by Student’s *t*-test (*n* = 3): * *p* < 0.05; ** *p* < 0.01; *** *p* < 0.001.

**Figure 5 microorganisms-08-01190-f005:**
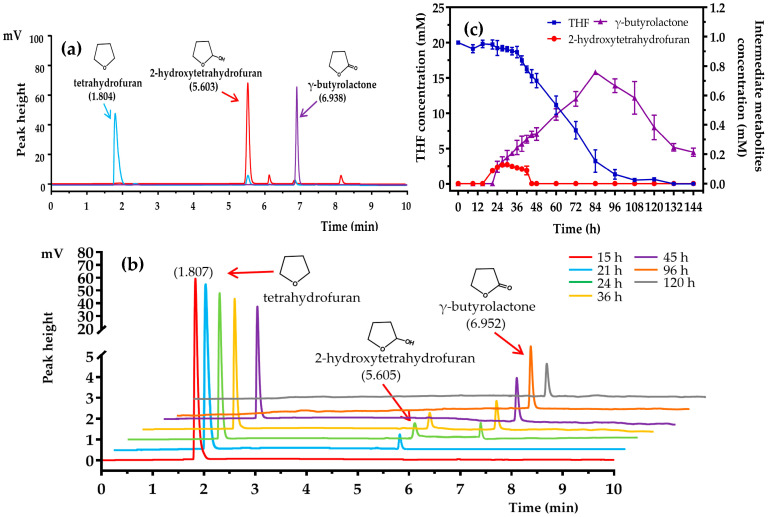
Detection and identification of metabolites during the biodegradation of 20 mM THF by strain ZM01 by GC analyses: (**a**) the retention time of different reference standards, including THF, 2-hydroxytetrahydrofuran, and γ-butyrolactone; (**b**) the detection and identification of the intermediate metabolites 2-hydroxytetrahydrofuran and γ-butyrolactone during the degradation of 20 mM THF by GC; and (**c**) the concentration changes of 2-hydroxytetrahydrofuran and γ-butyrolactone during the degradation of 20 mM THF by strain ZM01.

**Figure 6 microorganisms-08-01190-f006:**
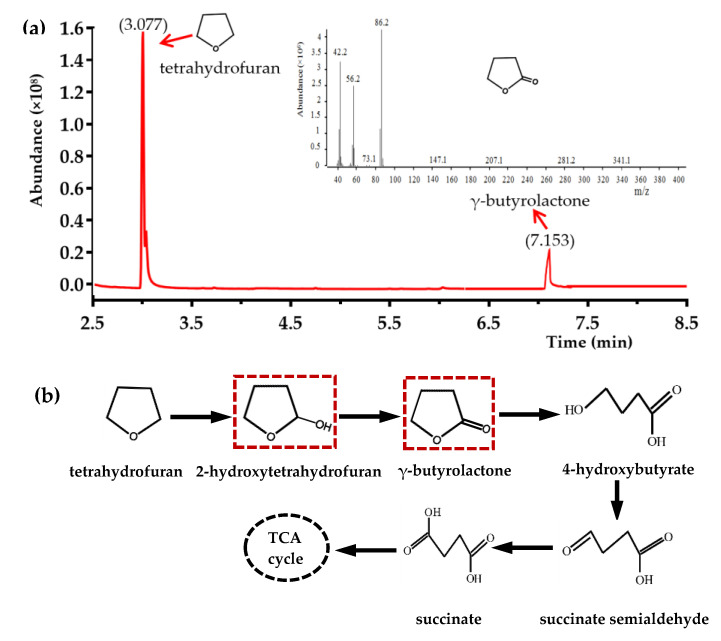
The identification of metabolites by GC-MS analyses and the proposed oxidation pathway of THF degradation by strain ZM01: (**a**) The MS spectrum of the produced metabolite γ-butyrolactone and (**b**) the presented pathway is based on the pathway originally suggested by Kristin, but the key metabolite 2-hydroxytetrahydrofuran was previously not been detected. The metabolites in the red dotted outline were detected during the THF degradation by strain ZM01. The actual formation of 2-hydroxytetrahydrofuran has been verified for the first time.

**Table 1 microorganisms-08-01190-t001:** Growth of strain ZM01 on different substrates.

Substrate	Growth ^a^ (mg)	Substrate	Growth ^a^ (mg)
Tetrahydrofuran	63.3 ± 2.5	1,4-Dioxane	8.0 ± 1.0
3-Hydroxytetrahydrofuran	8.0 ± 1.0	Pyridine	NG
Dimethylfuran	6.2 ± 0.5	Ethyl acetate	17.7 ± 1.5
γ-Butyrolactone	53.7 ± 1.5	Methyl alcohol	NG
1,4-Butanediol	71.7 ± 1.5	*n*-Hexane	8.7 ± 0.6
Benzene	8.3 ± 0.6	Acetone	NG
Methylbenzene	7.7 ± 0.6	2-hydroxytetrahydrofuran	58.9 ± 2.1

“^a^” indicates the growth (mycelial dry weight) of strain ZM01 on corresponding substrates. “NG” indicates no growth. Strain ZM01 was cultured with each type of substrate (20 mM) for 5 d at pH 7.0, 200 rpm, and 30 °C.

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
