# Peer review of "Biodegradation of Tetrahydrofuran by the Newly Isolated Filamentous Fungus Pseudallescheria boydii ZM01"

_microorganisms, 2020, doi:10.3390/microorganisms8081190_

Round 1

Reviewer 1 Report

Well written and interesting paper describing a versatile fungus capable of degrading relatively high concentrations of THF under a broad range of environmental conditions.  The one question that arose during reading was whether the fungus grows on 2-hydroxyTHF?  One minor comment, please give the unit for the mycelial dry weight in Table 1.

Author Response

Dear editor:

Thanks very much for your handling review of this paper. We also appreciate the insightful comments and suggestions of the two anonymous reviewers, which are very valuable for us to improve this manuscript. We have carefully considered each point raised by all the reviewers and attempted to answer the questions and explain our thoughts. We believe that the quality and readability of this manuscript have been significantly improved. By considering the reviewers’ comments, the following major changes have been made in the text. All the revisions have been clearly highlighted in Microsoft word using the “Track Changes” function.

Response to Reviewer 1 Comments

Point 1: Well written and interesting paper describing a versatile fungus capable of degrading relatively high concentrations of THF under a broad range of environmental conditions. The one question that arose during reading was whether the fungus grows on 2-hydroxyTHF? One minor comment, please give the unit for the mycelial dry weight in Table 1.

Response 1: Thank you very much for your comments. The THF-degrading filamentous fungus P. boydii ZM01 can utilize 2-hydroxytetrahydrofuran as a sole carbon source, and the mycelial dry weight was up to 58.9±2.1 mg after cultivation for 5 d at 30℃. The mycelial dry weight of strain ZM01 using 20 mM 2-hydroxytetrahydrofuran as a substrate has been added to the paper (line 387 and 389) and Table 1 (line 377). The unit for mycelial dry weight was “mg” and has been supplemented in Table 1 (line 376).

Reviewer 2 Report

The article is devoted to the isolation of a fungus capable of THF destruction and to the study of this process.

The material presented in the work is of interest for research. The article corresponds to the profile of the journal and can be published after making minor corrections.

There are some remarks:

Lines 75-77 it is written: “In the oxidation pathway, THF is inferred to be initially oxidized to 2-hydroxytetrahydrofuran, which is then gradually oxidized to γ-butyrolactone…”

is the process sequential or parallel? Since according to the data of the chemical oxidation of THF,  liquid-phase oxidation of O2 air in the presence peroxides, salts of transition metals or B2O3 proceeds with the formation of 2-hydroxytetrahydrofuran (yield 15-17%) and g-butyrolactone (75-80%).

Line 78 – mistake – “2-hydroxytetrahydrofura”

Lines 103-104 “Minimal salt medium (MSM) supplemented with THF was initially to isolate and culture THF degrading microorganisms” probably should be modifies as “Minimal salt medium (MSM) supplemented with THF was initially used to isolate and culture THF104 degrading microorganisms”

Lines 117-119. It is written “The collected samples were acclimatized in 100 mL of MSM supplemented with different THF to different final concentrations (from 20 mM to 50 mM) at 30°C with continuous shaking at 200 rpm.”

What kind of different THF was used? May be “different THF” should be replaced by “THF and its analogs”?

Lines 160. It is written that substrates were used at concentration 20 mM. It is rather high (for THF it equal to 1.4 g/L) concentration. Are all substrates to be soluble at this concentration? Was it any toxic effect?

Line 532.  Environ sci technol – all words for the title of the journal must begin with a capital letter.

Lines 605-606 – “ Curry, S.; Ciuffetti, L.; Hyman, M. Inhibition of growth of a Graphium sp. on gaseous n-alkanes by gaseous n-alkynes and n-alkenes. Appl Environ Microbiol. 1996, 62, 2198-2200.” “n” – italic.

I also have a comment to the caption of figure 3 of the supplementary materials «The degradation curve of strain ZM01 using THF and γ-butyrolactone as substrates. The red line represents growth with THF as the sole substrate, the purple line represents growth with γ-butyrolactone as the sole substrate, and the blue line represents growth with THF and γ-butyrolactone as substrates. The solid line represents the THF concentration change, and the dotted line represents the γ-butyrolactone concentration change.» Lines does not represent the growth with THF, they show the decrease of concentration of e.g. THF.  And in total, the lines presented do not reflect growth, but the process of destruction. The caption should be rephrased as “…The red line represents cultivation with THF as the sole substrate, the purple line represents cultivation…”

Author Response

Dear editor:

Thanks very much for your handling review of this paper. We also appreciate the insightful comments and suggestions of the two anonymous reviewers, which are very valuable for us to improve this manuscript. We have carefully considered each point raised by all the reviewers and attempted to answer the questions and explain our thoughts. We believe that the quality and readability of this manuscript have been significantly improved. By considering the reviewers’ comments, the following major changes have been made in the text. All the revisions have been clearly highlighted in Microsoft word using the “Track Changes” function.

Response to Reviewer 2 Comments

Point 1: Lines 75-77 it is written: “In the oxidation pathway, THF is inferred to be initially oxidized to 2-hydroxytetrahydrofuran, which is then gradually oxidized to γ-butyrolactone…” is the process sequential or parallel? Since according to the data of the chemical oxidation of THF, liquid-phase oxidation of O2 air in the presence peroxides, salts of transition metals or B2O3 proceeds with the formation of 2-hydroxytetrahydrofuran (yield 15-17%) and γ-butyrolactone (75-80%).

Response 1: Thank you. It’s an interesting process in the chemical oxidation of THF. To the best of our knowledge, the THF biodegradation process is sequential and different from the chemical oxidation process. To avoid misunderstanding, we have corrected the sentences in line 75-78 to “In the oxidation pathway, THF is inferred to be initially oxidized to 2-hydroxytetrahydrofuran, and 2-hydroxytetrahydrofuran is then oxidized to γ-butyrolactone before being transformed to 4-hydroxybutyrate”. The proposed THF biodegradation pathway is based on the studies in THF-degrading fungus Graphium sp. and THF-degrading bacterium Rhodococcus aetherivorans strain M8.

References

Tajima, T.; Hayashida, N.; Matsumura, R.; Omura, A.; Nakashimada, Y.; Kato, J. Isolation and characterization of tetrahydrofuran-degrading Rhodococcus aetherivorans strain M8. Process Biochemistry. 2012, 47, 1665-1669.

Skinner, K.; Cuiffetti, L.; Hyman, M. Metabolism and cometabolism of cyclic ethers by a filamentous fungus, a Graphium sp. Applied and Environmental Microbiology. 2009, 75, 5514-5522.

Point 2: Line 78-mistake-“2-hydroxytetrahydrofura”

Response 2: Thanks for pointing this spelling error out. We have corrected it to “2-hydroxytetrahydrofuran” in Line 78.

Point 3: Lines 103-104 “Minimal salt medium (MSM) supplemented with THF was initially to isolate and culture THF degrading microorganisms” probably should be modifies as “Minimal salt medium (MSM) supplemented with THF was initially used to isolate and culture THF degrading microorganisms.”

Response 3: Thank you very much for your comments. We have added the word “used” in this sentence (line 103) according to your comments.

Point 4: Lines 117-119. It is written “The collected samples were acclimatized in 100 mL of MSM supplemented with different THF to different final concentrations (from 20 mM to 50 mM) at 30°C with continuous shaking at 200 rpm.” What kind of different THF was used? May be “different THF” should be replaced by “THF and its analogs”?

Response 4: We are sorry for the unclear description. We want to describe the supplement of different concentrations of THF, not different kinds of THF or THF analogs. To avoid misunderstanding, this sentence has been corrected to “The collected samples were acclimatized in 100 mL of MSM supplemented with different concentrations of THF from 20 mM to 50 mM at 30°C with continuous shaking at 200 rpm” (line 117-119).

Point 5: Lines 160. It is written that substrates were used at concentration 20 mM. It is rather high (for THF it equal to 1.4 g/L) concentration. Are all substrates to be soluble at this concentration? Was it any toxic effect?

Response 5: Thanks for pointing this out. To the best of our knowledge, some of the substrates are insoluble at this concentration such as dimethylfuran, methylbenzene, dimethylbenzene, n-hexane, dichloromethane, and trichloromethane. P. boydii ZM01 can utilize the insoluble dimethylfuran, methylbenzene, and n-hexane as the sole carbon source. Considering the effect of water solubility of substrates on degradation, the results of strain ZM01 growing on dimethylbenzene, dichloromethane and trichloromethane were deleted in Table 1 (Line 376). In this study, we want to evaluate the degradation potential of P. boydii ZM01 under high concentration substrates stress. Thus, a rather high substrate concentration was used here even though it may be toxic to THF degrading strain.

Point 6: Line 532. Environ sci technol - all words for the title of the journal must begin with a capital letter.

Response 6: Thanks for pointing this out. We have checked all references and corrected the journal name in line 523 and line 589.

Point 7: Lines 605-606 “ Curry, S.; Ciuffetti, L.; Hyman, M. Inhibition of growth of a Graphium sp. on gaseous n-alkanes by gaseous n-alkynes and n-alkenes. Appl Environ Microbiol. 1996, 62, 2198-2200.” “n” – italic.

Response 7: Thank you. We have checked all this kind of format error in this paper and corrected them in Table 1, Line 388, and line 596-597.

Point 8: I also have a comment to the caption of figure 3 of the supplementary materials “The degradation curve of strain ZM01 using THF and γ-butyrolactone as substrates. The red line represents growth with THF as the sole substrate, the purple line represents growth with γ-butyrolactone as the sole substrate, and the blue line represents growth with THF and γ-butyrolactone as substrates. The solid line represents the THF concentration change, and the dotted line represents the γ-butyrolactone concentration change”. Lines does not represent the growth with THF, they show the decrease of concentration of e.g. THF. And in total, the lines presented do not reflect growth, but the process of destruction. The caption should be rephrased as “…The red line represents cultivation with THF as the sole substrate, the purple line represents cultivation…”

Response 8: Thanks for pointing this out. We are sorry for the wrong description in the caption of Figure S3 (line 707-709). We have rephrased the caption as “The red line represents cultivation with THF as the sole substrate, the purple line represents cultivation with γ-butyrolactone as the sole substrate, and the blue line represents cultivation with THF and γ-butyrolactone as substrates”.

Round 2

Reviewer 2 Report

The authors made corrections to the text of the article. The manuscript can be accepted for publication in the journal Microorganisms.

Author Response

Point 1: Since spores are made by this fungus - the authors should say something about how their morphology compares to that of the same fungus as shown in the literature. Sometimes simple ITS data do not fit the data from fungus morphology - the authors need to clarify this and it can be easily done... Otherwise the report is nicely done but other morphology work needs to be mentioned, etc.

Response 1: Thanks for your insightful comments and suggestions. We have shown the colony morphology and scanning electron microscopic photographs of strain ZM01 as Figure S1 in supplementary materials (Line 17-26). The morphology characteristics of strain ZM01 has been described and supplemented in the paper (Line 189-201) as follows: Strain ZM01 could grow as white thread-like hyphae and develop conidia growing directly on vegetative hyphae after cultivation for 7 d at 30℃. Most conidia were usually brown, thick-walled, globose to subglobose. The shape of conidiogenous cells was cylindrical, and no yellow diffusible pigment was produced during the cultivation of strain ZM01 on PDA at 30℃ (Figure S1). The microscopic characteristics of strain ZM01 were consistent with the typical microscopic features of Pseudallescheria sp.. Based on phylogenetic analysis of the ITS region sequences (GenBank Accession No. MT754398) (Figure 1), strain ZM01 was observed to be closely related to P. boydii (GenBank Accession No. AY213683) with a high degree of similarity (99%). Thus, on the basis of morphological characteristics and phylogenetic relationships analysis, we identified the isolated THF-degrading fungus as Pseudallescheria boydii strain ZM01. All the revisions have been clearly highlighted in Microsoft word using the “Track Changes” function.
